# The Role of Dietary and Microbial Fatty Acids in the Control of Inflammation in Neonatal Piglets

**DOI:** 10.3390/ani11102781

**Published:** 2021-09-24

**Authors:** Barbara U. Metzler-Zebeli

**Affiliations:** Unit Nutritional Physiology, Institute of Physiology, Pathophysiology and Biophysics, Department of Biomedical Sciences, University of Veterinary Medicine, Veterinaerplatz 1, 1210 Vienna, Austria; barbara.metzler@vetmeduni.ac.at

**Keywords:** autoxidation, colostrum, neonatal piglet, polyunsaturated fatty acids, maternal diet, porcine milk, microbial fatty acid derivatives, lipid mediators, inflammation

## Abstract

**Simple Summary:**

The maturation of the gut is a specific and very dynamic process in new-born piglets. Consequently, piglet’s gut is very susceptible to disturbances, especially in stressful periods of life, such as weaning, when the gut lining often becomes inflamed and leaky. Dietary fatty acids (FA) do not only serve as source of energy and essential FA, but they are important precursors for bioactive lipid mediators, which modulate inflammatory signalling in the body. The current review summarizes results on dietary sources of FA for piglets, the signalling cascades, bioactivities, the necessity to consider the autoxidation potential of polyunsaturated FA and the area of microbially produced long-chain FA. That said, porcine milk is high in fat, whereby the milk FA composition partly depends on the dietary FA composition of the sow. Therefore, manipulation of the sow diet is an efficient tool to increase the piglet’s intake of specific FA, e.g., n-3 polyunsaturated FA which show anti-inflammatory activity and may support intestinal integrity and functioning in the growing animal.

**Abstract:**

Excessive inflammation and a reduced gut mucosal barrier are major causes for gut dysfunction in piglets. The fatty acid (FA) composition of the membrane lipids is crucial for mediating inflammatory signalling and is largely determined by their dietary intake. Porcine colostrum and milk are the major sources of fat in neonatal piglets. Both are rich in fat, demonstrating the dependence of the young metabolism from fat and providing the young organism with the optimum profile of lipids for growth and development. The manipulation of sow’s dietary polyunsaturated FA (PUFA) intake has been shown to be an efficient strategy to increase the transfer of specific FAs to the piglet for incorporation in enteric tissues and cell membranes. n-3 PUFAs, especially seems to be beneficial for the immune response and gut epithelial barrier function, supporting the piglet’s enteric defences in situations of increased stress such as weaning. Little is known about microbial lipid mediators and their role in gut barrier function and inhibition of inflammation in neonatal piglets. The present review summarizes the current knowledge of lipid nutrition in new-born piglets, comparing the FA ingestion from milk and plant-based lipid sources and touching the areas of host lipid signalling, inflammatory signalling and microbially derived FAs.

## 1. Introduction

In pig nutrition, lipids are principally considered as a source of energy and essential fatty acids [FA; i.e., alpha-linolenic acid, (ALA, 18:3n-3) and linoleic acid (LA 18:2n-6)] [1]. Post farrowing, porcine colostrum and milk are the major source of fat for the piglets during the first weeks of life. Porcine milk is very rich in lipids, demonstrating the dependence of the neonatal metabolism on fat and providing the young organisms with the optimum profile of lipids comprising triglycerides, phospholipids and cholesterol. Since dietary FA are directly utilized by monogastric animals, the FA profile of sow colostrum and milk reflects her dietary intake, providing an opportunity to modify the FA composition of the milk and the FA supply of the piglet. Therefore, the lipid content and FA profile of the sow milk should be used as a reference when formulating milk replacers and prestarter and starter diets for piglets. However, the fat sources utilized in these feeds comprise a different FA composition. Hence, the FA profile of milk replacers corresponds to the main fats, most often coconut oil and palm oil. Prestarter and starter diets contain only small amounts of fat deriving mostly from cereals, protein-rich by-products and added processed vegetable oils which contain higher concentrations of polyunsaturated FAs (PUFA) and wider omega-6 (n-6) PUFA to omega-3 (n-3) PUFA ratios, fewer saturated FA and no cholesterol compared to porcine milk. Due to the important roles that certain FAs play in relation to mucosal immune responses, epithelial barrier function, oxidative stress, and inflammatory reactions, more attention has been paid lately towards their provision of neonatal and weaned piglets [1].

In general, the term “lipids” comprises a number of hydrophobic compounds, including triglycerides, phospholipids, sphingolipids, cholesterol and ester derivatives, and fat-soluble vitamins. They also include the single FAs (C6 to C32) and their derivatives, as well as related functional metabolites derived from them [2]. These compounds have biological activities that act to influence the function and responsiveness of cell membranes and tissue metabolism, to hormonal and immune signalling. FAs exists in two main forms, saturated or unsaturated, containing one or more double bonds; the latter being of greater physiological importance because of their reactivity towards oxygen and bioactive signalling. Mammals do not express Δ12-desaturase and Δ15-desaturase-enzymes which are required for the production of ALA and LA [3]. Therefore, n-3 and n-6 PUFAs need to be obtained by the diet and the balance between n-3 and n-6 PUFAs depends on the composition of the dietary lipids consumed [4]. *De novo* FA synthesis of long-chain PUFAs through elongation and desaturation is a saturable process. The animal may therefore benefit from supplementation of long-chain PUFAs to compensate for limited production in a situation of increased demand. Of particular interest for inflammatory signalling are the unsaturated long-chain n-3 and n-6 PUFAs. Especially, the long-chain PUFAs eicosapentaenoic acid (EPA, C20:5 n-3), docosahexaenoic acid (DHA, 22:6n-3 n-3) and arachidonic acid (AA, C20:4 n-6) play crucial roles for neonatal development, including brain development, health and immunity. Other fat types important for the membranous phospholipid integrity and fluidity, such as cholesterol, or saturated [5], have been poorly researched in relation to gut integrity and function in traditional pig nutrition so far.

Feeding the mother sow divergent ratios of n-6:n-3 PUFA can alter not only their health, physiological condition, and reproductive performance but also exert similar effects in their offspring. These findings for pigs are supported by observations made in humans where a strong link between the intake of n-6 PUFAs and the incidences of (gut) inflammatory disorders exist [6]. While for humans an optimal n-6:n-3 ratio of 2 to 3:1 has been suggested, various n-6:n-3 ratios have been proposed in pig nutrition, but ratios may differ for pigs of different ages and the search for an optimal proportion in the feed for sows and piglets is still in progress [7]. Based on this information, more knowledge about the role of FAs in the early neonatal nutrition of pigs in relation to the maturation of the gut and immune system needs to be gained.

While a great number of studies described the impact of supplementing different fat sources for sow diets during the late gestation and lactating period with regard to the performance of sows and their progeny [1], our knowledge on the potential of dietary FAs, especially PUFAs and their bioactive abilities in neonatal and weaned piglets is still rising. With our advancing understanding of the critical impact of dietary lipids in the modulation of the intestinal immune response, the aim of this literature review is to provide an overview of the current knowledge of lipid nutrition in the early life of pigs, focusing on the changes in FA ingestion from the suckling to the postweaning period by comparing milk fat and plant-based lipid sources used in solid diets, host lipid signalling, inflammatory pathways triggered by PUFA and the area of microbially derived long-chain -FAs. Short-chain fatty acids (SCFA) are a major part of the FAs produced by the microbiota in intestinal digesta with many beneficial effects for the host. However, these will be only briefly presented in the present review.

## 2. Role of Lipids for Piglet’s Development

Piglets from modern hyperprolifc hybrid lines gain weight quickly, doubling their body weight within the first week of life, and reach a weight of about 5- to 7-times their birth weight at weaning with 4 weeks of age. Consequently, the developing organs and tissues require high amounts of energy and nutrients for growth and proper functioning. Especially the long-chain EPA and DHA play crucial roles for the prenatal and postnatal development of the brain and retina, as well as in the development of proper immunity in new-borns [8,9]. The dietary FA profile largely determines the lipid composition of membranes, including the gut epithelium, which has consequences for epithelial integrity and functioning. The maturation of the gut is a specific and very dynamic process in new-borns. It is influenced by intrinsic (i.e., genotype) and extrinsic factors, including the first nutrition and the developing gut microbiota. As a consequence, the developing gut epithelium is continuously exposed to alterations in nutritional, microbial, environmental and physiological factors during the first weeks of life [10,11].

Small size piglets, which have become common due to the large litter sizes of modern hyperprolific sows [1,12], show signs of compromised gut integrity and functionality and may benefit from extra nutrition. As another critical time point in the life of a piglet, the increased stress experienced at weaning activates the inflammatory response which often results in the degeneration of the gut epithelial structure due to anorexia, gut inflammation and psychological stressors [13,14]. This additionally impacts the still immature digestive and absorptive capacity, gut mucosal integrity and animal health, allowing the translocation of pathogens and toxins to the systemic circulation [13]. Especially, when the piglet starts eating again, undigested material remains in the gut, which provides good conditions for bacterial pathogens to proliferate [14]. Feeding a ration with low inflammatory potential is therefore mandatory to reduce stressors for gut mucosal functioning. The composition of dietary lipids greatly changes from the suckling to postweaning phase, depending on the extent to which the piglet has access to milk replacer and commercial piglet feed preweaning. In the suckling phase, lipids, including PUFAs, saturated fats and cholesterol, are provided in necessary amounts by mother’s milk. By contrast, porcine solid diets (i.e., creep feed and starter diets) comprise in general less fat and saturated FAs than sow milk. Simultaneously, the solid feed contains higher concentrations of PUFAs and wider n-6:n-3 PUFA ratios and no cholesterol compared to porcine milk. Therefore, in terms of the lipid and FA profiles, piglets may be exposed to more proinflammatory stimuli preweaning than foreseen by nature at an age of 28 days. While the supplementation level and type of the dietary lipid source can influence the effects of n-3 PUFAs, the dietary supplementation of the diets for sows beginning in late gestation and during lactation seem to be an efficient option to increase the n-3 PUFAs in piglets, as will be outlined in the following paragraphs. In this respect, certain evidence exits that supplementation of sow’s diets or the (pre-)starter diet of the piglets with n-3 PUFA rich oils, such as fish oil, linseed oil or hempseed oil, can increase the n-3:n-6 PUFA ratio in their offspring, which may improve the supply with immunoglobulins, attenuate the inflammatory response in the gut and systemically reduce the acute physiological stress response in the pigs postweaning [15,16,17,18]. Against this background, both low birth weight and newly weaned piglets may benefit from potential anti-inflammatory properties of n-3 PUFAs. Moreover, the hepatic conversion of the essential n-3 ALA and n-6 LA into their long-chain bioactive counterparts are saturable processes, which may limit their adequate provision in situations of increased demand. Therefore, the question is valid whether the dietary lipid composition and amounts provided in commercial starter feeds meet piglet’s requirements in situations of increased demand.

## 3. Lipids as Source of Energy and Nutrients

Porcine colostrum, transient and mature milk contain a complex mixture of nutritional, bioactive and immunological compounds, including carbohydrates (i.e., lactose and milk oligosaccharides), lipids, proteins, biogenic amines, and growth factors [19], and naturally constitute the main source of nutrition for the new-born piglet. While fluctuating during the course of lactation, the nutrient concentrations and proportions in colostrum and milk reflect the nutrient needs of the new-born and stimulate maturational processes and growth. Late colostrum and transient milk are especially rich in fat [20], demonstrating the importance of this nutrient for piglet’s metabolism. In fact, during postpartum, the metabolism of piglets shifts from being glucogenic to becoming ketogenic [21]. That means that colostral and milk fat are the main energy sources of the new-born piglet while providing all essential FAs. The high milk fat content is matched by the secretion of potent lingual, gastric and pancreatic lipases, ensuring maximum utilization of the colostral fat by rendering it highly digestible [22]. In that way, milk fat fulfils a dual role by providing insulation and an energetic substrate for thermoregulation via the hepatic oxidation of fat to CO_2_. The liver of newborn piglets can completely oxidize medium and long-chain FA, whereas the glycogen stored in the liver is used as glucogenic substrates to fuel oxidation of ketogenic (fat) energy [23].

### 3.1. Lipid Digestion and Absorption

The digestion of dietary lipids along the gastrointestinal tract has been described recently and the reader is referred to these reviews [1,24]. The neonatal piglet has a high ability to digest milk fat, and it can be assumed that fat digestion and absorption from plant sources is less efficient relative to older pigs [25]. This is mainly due to the involvement of gastric lipases in neonates, which is inferior to the pancreatic lipase activity (only 3%), but it may already hydrolyse 25 to 50% of triglycerides in the stomach [26]. Whether neonatal piglets mainly raised on milk replacer have a reduced fat digestion and absorption as described for human formula-fed infants relative to infants fed mother’s milk [27] can be speculated. In this respect, the chain length and degree of saturation of FAs as well as their positional distribution in triglycerides are important determinants for fat digestion and absorption [26], as they are differently absorbed and metabolized, including active and passive routes.

After absorption from the intestinal lumen, which is illustrated in detail for pigs elsewhere [1,24], the majority of long-chain FAs are transported in chylomicrons via the lymphatic duct to the liver for metabolism and packaging into lipoproteins. In addition to being a source of energy, long-chain PUFAs have important structural functions as part of the phospholipid bilayer in cell membranes, affecting membrane fluidity, and intracellular signal transduction mechanisms [28]. The latter influences many important physiological processes, including lipid mediator production, cell division, immune function, inflammation and cell differentiation [24].

### 3.2. Gut Mucosal Recognition of FAs

The biological function of FAs partly depends on their chain length and systemic availability to be used by the body for lipid synthesis. Most of the research about intestinal lipid and FA signalling in pigs has been conducted for the postweaning phase, leaving a gap of knowledge for the early neonatal period. Before absorption, the free FAs in intestinal digesta is recognized by mucosal receptors, which systematically regulate physiological processes at the mucosa. At the apical membranes of the enterocyte, the FA bind to G-protein-coupled receptors [GPRs; also called free fatty acid receptors (FFARs)], hydroxycaboxylic acid receptors (HCARs) and nuclear transcription factors, such as peroxisome proliferator activated receptors (PPARs) [29]. These receptors are expressed along the gastrointestinal tract and are also found on hepatocytes, immune cells and adipocytes. In addition, other less well described mechanisms seem to mediate biological effects of FAs at the gut mucosa [29]. The biological functions and antigens have been characterized only for a small number of the estimated 800 GPRs; especially data for farm animals are still limited [30]. Consequently, very little has been described for the functional importance of these receptor families in neonatal piglets. The GPRs are selective for a particular free FA carbon chain length derived from food or food-derived metabolites, among those FFAR-1 (GPR40) and FFAR-4 (GPR120), activated by medium-chain FAs (MCFA; GPR-40) and long-chain FAs (GPR-40 and GPR-120), as well as FFAR-2 (GPR43) and FFAR-3 (GPR41) which are mainly activated by SCFAs [29]. GPRs, HCARs and nuclear transcription factors are involved in signalling related to insulin and incretin hormone secretion, adipocyte differentiation, anti-inflammatory effects, neuronal responses, and appetite and thus are involved in energy and immune homeostasis. Recently, we have been able to demonstrate the regional specificities for gut mucosal SCFA, lactate and long-chain FA sensing capacities (i.e., receptor activation and FA signal transduction) in the small and large intestines of 10-week-old pigs towards luminal FA concentrations, leading to the differential expression of the downstream targets of these receptors, i.e., gastric inhibitory polypeptide (GIP) and glucagon-like peptide (GLP)-1, in the small and large intestines [31]. At this age, GPRs and HCAR-1, which responds to lactate, were highly expressed in the small and large intestines, emphasising the importance of dietary and microbial FA sensing along the intestinal tract. Gene expression data further suggested a stronger signalling via FFAR-1 in the ileum, as well as via FFAR-2, FFAR-4 and HCAR-1 in the cecum and colon regions which was likely attributable to substrate affinities and luminal exposure to SCFA, MCFA and long-chain FA [31]. Likewise, gene copy abundances for nuclear transcription factors suggested a greater role of FA synthase (FASN) and sterol regulatory element-binding protein (SREBP)-2 in gut mucosal signalling in young growing pigs [31]. However, information about the development of gut region-specific FA receptor expression and signalling in the first weeks of life is still missing. Porcine milk contains a broad profile of saturated and unsaturated FAs of various chain length; therefore, respective expression profiles may be assumed. Unpublished data from our group indicate jejunal and cecal abundances of GPRs and HCAR-1 of 3 to 6 log_10_ gene copy numbers/g tissue from day 6 to day 34 of life (weaning at day 28 of life), which supports our assumption [Lerch, Vötterl, Koger, Metzler-Zebeli; personal communication]. Effects of SCFAs, such as butyrate, on the colonic neuronal excitability and hence colonic motility via increased histone H3 acetylation in enteric neurons of rats has been described [32]. However, there is little information on the regulatory role of FAs on the development of the neuronal signalling in the gut of neonatal piglets.

### 3.3. Physiological Roles of PUFAs

Omega-3 and n-6 PUFAs can influence the inflammatory state of a range of cell types, including epithelial and immune cells [6]. When integrated into the phospholipid bilayer of membranes such as the gut epithelium, n-3 and n-6 PUFAs act as a reservoir of bioactive molecules and regulate the cell signalling pathways related to inflammation. The composition of PUFAs in porcine colostrum and milk is influenced by the FA profile of the sow diet, comprising the n-3 PUFAs ALA (C18:3), eicosadienoic acid (C20:2), EPA (C20:5), and DHA (C22:6) and n-6 PUFAs LA (18:3), γ-LA (18:3), eicosatrienoic acid (C20:3), AA (20:4). The main sources of n-3 and n-6 PUFA in solid diets for piglets are vegetable oils, in which mainly the essential n-3 PUFA ALA and the essential n-6 PUFA LA are present. While LA and ALA need to be converted into bioactive long-chain PUFAs, the longer chain PUFAs act as direct precursors for the synthesis of bioactive lipid mediators. Most research on potent lipid mediators have been done using cell cultures and the physiological functions of PUFA-derived lipid mediators are only well understood for a limited number of PUFAs, mainly AA, EPA and DHA [2,4]. Roles for other long-chain PUFAs such as n-3 docosapentaenoic acid (C22:5) are now emerging. Eicosanoids generated from AA represent one of the most potent classes of endogenous proinflammatory mediators, whereas resolvins, protectins and maresins derived from n-3 PUFAs exert anti-allergic and anti-inflammatory responses [2,33,34]. However, it should not be overlooked that AA-derived lipid mediators have both pro- and anti-inflammatory activities in the intestine and that AA-derived eicosanoids play valid roles in cell signalling, whereby their effect seems to be determined by the target cell type and receptor cell type [4]. Due to the production of inflammatory signals, the membrane becomes depleted of the respective long-chain PUFAs. This gap needs to be refilled by dietary or de novo synthesized AA from LA or related PUFAs to maintain the membranous integrity. During inflammation, the cytokine signalling cascade initiates the release of AA, EPA, DHA or related PUFAs from the phospholipid bilayers [2]. AA is converted by cyclooxygenases (COX1 and COX2), lipoxygenases, cytochrome p450, and other downstream enzymes, such as soluble epoxide hydrolase and non-enzymatic reactions into prostaglandins (PGs), leukotrienes (LTs), thromboxanes, lipoxins, hydroxyeicosatetraenoic acids and epoxyeicosatrienoic acids [2,4]. In cell culture models, AA products, such as PGE_2_ and LTB_4_, were more powerful inflammatory stimuli than EPA-derived PGE_3_ and LTB_5_, which exert anti-inflammatory activities [2]. Taken together, the proposed mechanisms by which EPA and DHA exert anti-inflammatory effects in the epithelium include partial replacement of AA in cellular phospholipids, thereby modifying the inflammatory signalling pathways through lipid mediator production. Simultaneously, they activate PPAR and GPR-related signalling which inhibit the action of the proinflammatory transcription factor nuclear factor kappa-light-chain-enhancer of activated B (NF-κB) [6]. As most of this information has been gained in several cell cultures and a rodent model, tackling the PUFA metabolite signalling cascade still needs to be confirmed in neonatal piglets.

### 3.4. Lipid Peroxidation

Irrespective of the type, PUFAs as constituents of cell membranes and in food have the unwanted feature to readily undergo non-enzymatic oxidation to form chemically-reactive species, especially when exposed to heat and light, due to the high reactivity of the double bonds [35,36,37,38]. Although this is a part of a normal process, oxidative damage caused by the increased abundance of free radicals is considered an important trigger factor in the pathogenesis of gut inflammatory diseases, compromising the epithelial barrier. Moreover, the conditions in the various segments of the gastrointestinal tract may accelerate the peroxidation of lipids and other dietary constituents, depending on the intestinal availability of antioxidants. Especially the conditions in the stomach (i.e., temperature, presence of catalysts and acid pH) may promote lipid peroxidation [35,36]. As a consequence, these pro-oxidant gastric conditions may reduce the PUFA bioavailability, increase PUFA peroxidation and limit the health benefits associated with dietary PUFAs (e.g., DHA and EPA). As described above, lipid bilayers are rich in PUFAs, e.g., LA and ALA, which are present in 80% of the total phospholipids. Consequently, membranes of the gut epithelium are very susceptible to reactive oxygen species (ROS)-initiated non-enzymatic lipid peroxidation of PUFAs, which alters membrane fluidity and function of membrane-bound enzymes, receptors and transporters, but also redox-sensitive signalling pathways and transcription factor expressions that sustain mucosal inflammation [35,36,37,38]. ROS impair the generation of ATP as PUFAs are essential for the optimal function of the respiratory electron transport and oxidative phosphorylation in mitochondria [39]. PUFA autoxidation is generally initiated by the abstraction of bis-allylic hydrogen atoms, such as, for example, hydrogen at position 11 for LA and at position 11 or 14 for ALA [38]. The rate of non-enzymatic, ROS-driven peroxidation of PUFAs thereby increases with the degree of unsaturation of the FA [3,25,40]. In addition to deteriorating membrane fluidity, oxidized PUFAs pass oxidative damage to other biomolecules, including proteins, through reactive carbonyl compounds, e.g., malondialdehyde, 4-hydroxy-nonenal, and 4-hydroxy-hexenal [38,39,40]. Therefore, lipid oxidation can have significant downstream effects and possibly play a major role in cell signalling pathways, triggered at the gut mucosa or after assimilation into the epithelial cell.

During the suckling phase, porcine milk is probably the freshest food the piglet can consume. Consequently, consumption of solid feed with the PUFA-rich vegetable oil and ground cereals (like corn or wheat) may present the greater risk to ingest oxidized FA. However, the FA profile of porcine milk reflects the dietary FA composition [41]. Therefore, the quality of the fat in sow’s gestation and lactation diets dictates the quality and oxidation stage of the FA ingested by the piglet during the suckling phase. In this respect, Su et al. [42] demonstrated that sows fed a diet with oxidized corn oil had an inferior milk quality and systemic oxidative status, as indicated by reduced serum activities of superoxide dismutase, compared to sows fed non-oxidized corn oil, which could be partially improved by supplementation of a commercial antioxidant (comprising ethoxyquin, citric acid and tertiary butyl hydroquinone). Therefore, piglets can experience the detrimental effects of deteriorized FAs without having direct access to the actual rancid fat source. This highlights the necessity to adequately store sow feed during times of summer heat and areas with warm climates and check the amount of antioxidants present in the feed on a regular basis. Moreover, supplements comprising PUFAs and linseeds, which are used to prevent obstipation in sows before farrowing, should not be fed beyond their expiration date.

## 4. Lipid Sources from Birth to Postweaning

Today’s pig diets are mostly plant-based, containing, if at all, few animal-based feedstuffs such as bovine dairy (by-)products or fish meal. In most complete feeds and single feedstuffs, the majority of dietary lipids is present as triglycerides, amounting to 95% of the dietary lipid fraction [1]. Other smaller fractions of dietary lipids include phospholipids, mainly as lecithin, cholesterol and cholesteryl esters if animal fat is present. Nevertheless, most phospholipids and cholesterol detected in the digestive tract originate from hepatic de novo synthesis and are secreted into the intestinal lumen with the bile. SCFAs and certain MCFAs, e.g., lactate and succinate, predominantly originate from gastrointestinal fermentation of (non-)digestible carbohydrates, including starch and fibre and contribute to the energy supply of the animal [1]. However, due to their beneficial effects, SCFAs, especially butyrate as sodium salt, are added as feed additives to pre-starter and starter diets.

Due to their importance for cognitive development, vision, cell metabolism, mucosal integrity and immune functions, PUFAs are specifically important for foetal and early neonatal development. In this context, the FA profile of new-born piglets reflects the maternal FA intake during pregnancy and lactation [18]. The foetus depends on the placental transfer of n-3 and n-6 PUFAs via maternal circulation [43,44]. Epitheliochorial types of placenta seem to have a limited capacity to transfer essential FAs [45]. Nevertheless, maternal plasma long-chain PUFAs start accumulating in larger amounts in the brain in the third trimester of pregnancy. Postnatally, they are provided by the colostrum and milk, whereby the actual concentrations change during the transition from colostrum to mature milk [43].

### 4.1. Lipid Composition of Porcine Colostrum and Milk

Colostrum and milk are the major food sources of piglets during their first days of life. On average, the fat content in mature porcine milk ranges from 6.5 to 8.6% fat, depending on the dietary fat level, breed and day of lactation of the sow [16,19,46,47] (Table 1). Colostrum contains less, while transition milk is richer in fat amounting to, on average, 5.6 and 8.1% fat, respectively [7,19,48]. Since the colostrum composition changes with progressing time postpartum, variations between studies can be related to the actual time point of the collection, the sow’s diet, breed and number of parities [48]. Using an endemic breed, Aguinaga et al. [49] showed for Iberian sows, an increase in milk fat from 3.8% in colostrum to 6.3, 6.1, 6.2, 5.4, and 5.6 on lactation days 5, 12, 19, 26 and 34, respectively. By contrast, in Large White × Landrace sows, the concentration of milk fat was higher in colostrum and milk, amounting to 6.2, 8.5, 9.0, 7.0 and 6.9 on lactation days 0 (colostrum), 7, 14, 21 and 28, respectively.

The majority of lipids in porcine colostrum and milk are triacylglycerols, followed by smaller levels of diacylglyerols, non-esterified FAs and cholesterol [48] (Table 1).

Recently, Suarez-Trujillo et al. [49] demonstrated the maturational changes in the lipid fractions of colostrum, transitional and mature milk. Specifically, mature milk contained higher levels of triacylglycerides and phosphatidylglycerols compared to colostrum (day 0) and transitional (days 3 and 7) and mature milk (day 14) [50]. Moreover, the carbon chain length and the unsaturation within the fatty acyl residues decreased from day 0 to 14 in both triglycerides and phosphatidylglycerols in their study [50]. Against popular belief, porcine colostrum and milk consist only to about 30% of saturated medium-chain FAs, whereas the other two-thirds consists of 30% of monounsaturated long-chain FAs and 30% of PUFAs [47,48]. The main FAs are palmitic acid (C16:0), oleic acid (C18:1) and LA (C18:2), representing about 80% of the FAs in porcine colostrum and milk [48,49,50,51]. Notably, the colostrum contains higher proportions of C18:2 n-6, C20:4 n-6, C20:5 n-3, C22:5 n-3, C22:6 n-3 than transition and mature milk [49], supporting their importance for early postnatal development of the piglets.

Regarding the other main lipid classes, cholesterol was the main sterol found in swine colostrum (~96% of total sterols) in the study of Luise et al. [48], while traces of sitosterol and campesterol have been also reported for porcine milk and milk from other species [51,52]. Moreover, porcine milk fat contains SCFA at barely detectable levels [50], thereby mirroring their concentration in serum. Porcine milk stimulates fermentation and thus increases SCFA and certain MCFA in the gastrointestinal tract of piglets due to their large amounts of disaccharides and oligosaccharides [53]. The lactose in the milk promotes lactate fermentation, commencing in piglet’s stomach, whereas milk oligosaccharides are important for early SCFA production [54].

Similar to other production stages, sow’s gestation and lactation diets are low in fat (2.5 to 5% ether extracts on as-fed basis), which mostly originate from cereals and n-6 PUFA rich vegetable oils [55]. Differences in lipid and FA composition of colostrum have been reported for the breed of sow and parity order [48]. For instance, Italian Duroc sows fed diets containing soybean oil as the main fat source secreted colostrum with 28% more fat, and 15% more C18:2 n-6, but 29% less C18:3 n-3 compared to Italian Large White sows [48]. With respect to parity, Large White sows in their fourth parity contained less oleic acid (C18:1 n-9) in colostrum, whereas their milk was richer in C18:2 n-6 PUFAs [48]. While the breed of sow is usually controlled in studies, the parity may impact the study outcome if not balanced among study groups. The most influencing factor for the colostrum and milk FA composition is undoubtedly the dietary FA profile, giving an opportunity to specifically increase the long-chain PUFA levels. Due to the ability of monogastric to directly utilize dietary FAs for body fat synthesis, the actual FA composition of porcine milk and colostrum changes with the FA composition of the fat sources of sow’s diet [56]. Lipids are usually added to sow diets due to their high energy availability compared to cereals and 6.0 g/day of the LA (C18:2 n-6) have been recommended for lactating sows in the past [55]. Due to the increasing awareness about anti- and proinflammatory properties of lipid mediators produced from n-3 and n-6 PUFAs, there is a certain effort to lower the ratio of n-6:n-3 PUFAs in maternal diets to below 10:1, for instance by using fish oil, hempseed oil or linseed oil to promote the health and performance of sows and piglets [57]. Fish oil is a good source for DHA and EPA, but unfortunately, it is not the most sustainable source of bioactive n-3 PUFAs. By contrast, seaweed, hempseed oil and linseed oil have become more and more accessible and thus are attractive sources of long-chain n-3 PUFAs and ALA, respectively. In addition, the feeding level of the sow in the week prior to farrowing influences the milk FA composition, comprising more n-3 PUFAs in sows receiving 4.5 kg feed per day compared to those receiving only 1.5 kg feed as their daily ration, as less fat is used as a source of energy [58]. We can assume that feeding levels differed among studies due to general practice at the experimental farm, breed, nutritional recommendation guidelines used in different countries and the actual need of the sows to reach optimal body condition, thereby contributing to the inter-study variation on effects of FAs on sow and piglet performance and physiology. This adds to the variation in the dietary fat level and the supplementation level of the fat source of interest, leaving more leeway for variation among studies.

Supplementation of gestation and lactation diets with fats rich in long-chain FAs to lactating sow diets did not only increase the colostrum and milk fat and the long-chain FA content in milk, but also altered the mono-unsaturated FA content and the n-6:n-3 PUFA ratio [46,47,59]. For instance, Bai et al. [46] compared MCFA-rich palm and coconut oil with n-6 PUFA rich soybean oil. The authors observed that the dietary inclusion of 3% soybean oil increased the colostrum PUFA concentration by 30%, especially the C18:2 n-6 and C18:3 n-3 fractions, whereas the milk of sows fed with 3% coconut oil and palm oil comprised higher level of lauric (C12:0), and myristic acid (C14:0) and palmitic acid (C16:0), respectively [46]. De Quelen et al. [59] compared n-6 and n-3 PUFA rich oils by feeding sows diets with either sunflower oil (low 18:3 n-3, with 18:3 n-3 representing 3% of total fatty acids) or a mixture of extruded linseed and sunflower oil (medium 18:3 n-3, with 9% of 18:3 n-3) or extruded linseed (high 18:3 n-3 with 27% of 18:3 n-3) during gestation and lactation. Results for the milk fat showed an enrichment with n-3 PUFAs in the milk from high 18:3 n-3 sows compared to that of low 18:3 n-3 and medium 18:3 n-3 sows [59]. Likewise, Nguyen et al. [7] lowered the n-6:n-3 PUFA ratio of the maternal diet by replacing soybean oil by linseed oil, resulting in a n-6:n-3 PUFA ratio of 13:1 for the control diet during gestation and 10:1 during lactation or the low n-6:n-3 ratio of 4:1 during gestation and lactation. As could have been expected from the aforementioned, the low n-6:n-3 ratio in the maternal diet reduced concentrations of γ-linolenic acid (GLA, C18:3 n-6) and DHA by two-fold and the overall n-6:n-3 ration by one-third in colostrum. Moreover, mature milk on day 7 of lactation was enriched in total n-3 PUFAs and ALA (C18:3 n-3), eicosatrienoic acid (ETA, C20:3 n-3), and also the increased level of eicosapentaenoic acid (EPA; C20:5 n-3) in sows fed the low n-6:n-3 PUFA diet to the same extent [7]. Overall, n-6:n-3 ratio was two-thirds declined in the low ratio treatment in their study. The findings of de Quelen et al. [59] and more recently of Nguyen et al. [7] emphasize the importance that the FA composition of the gestation diet has on the FA profile of colostrum, providing not only essential FA for the early neonatal development but also influencing the inflammatory signalling in the new-born. Therefore, the quality of colostrum should be assessed in terms of the immunoglobulin concentrations and additionally, by the concentrations of other bioactive nutrients, such as FA, are integrated in this assessment. During prepartum, the effect of lowering the n-6:n-3 ratio seemed to be less effective. Supplementation of n-3 PUFA during gestation did not provide benefits for embryo survival and no consistent effects on litter performance during lactation were found, as reviewed by Tange and de Smet [28]. Based on these findings, it seems more appropriate to begin with the supplementation of the sows with n-3 PUFA rich oils in late gestation. However, more research is needed to address the factors of overall dietary fat content and feed intake level as interactive factors for the outcome of lowering n-6:n-3 PUFA ratio of sow’s diet.

The physiological processes which aid the FAs are taken up by the mammary gland and how they are utilized for milk fat synthesis has been described elsewhere [19,28]. Only that much, during the de novo synthesis of lipids in the mammary gland, lysophosphatidic acid is required for forming the milk fat and influencing the FA profile [60]. It is one of the mediators in bioactive LPL [61]. In this respect, the observations made by Jang et al. [62] suggest that exogenous LPL, when included in sow’s diet during lactation, may be used to modify the de novo pathway for lipid synthesis in the mammary gland, which may be used to specifically alter the FA composition of porcine milk. In their study, sows fed diets with LPL had increased ratios of n-6:n-3 PUFAs and unsaturated/saturated FAs with decreased myristic acid (C14:0), palmitic acid (C16:0) and increased oleic acid (C18:1 n-9), LA (C18:2 n-3), eicosenoic acid (C20:1 n-11), and eicosadienoic acid (C20:2 n-6) in the milk during lactation.

Taken together, porcine colostrum and milk are the most natural sources to provide new-born piglets with various types and ratios of lipids necessary for proper development and with beneficial bioactive properties. Sow’s milk can be enriched with specific bioactive FA, which can be achieved by appropriate formulation of sow’s gestation and lactation diets.

### 4.2. Lipids and Bioactive FA in Milk Replacers and Solid Diets

Nursing by the sow is often complemented with plant-based milk replacers to account for big litter sizes in commercial settings as early as day 2 of life [63]. Especially when the number of piglets surpasses the number of functional teats, offering milk replacer allows the surplus piglets to stay with their own mother, while providing extra nutrition and a certain training of the intestine to solid feed components preweaning. Milk replacers, creep feed, and pre-starter diets are formulated based on economic aspects and macro-nutrients. They often lack functional milk compounds, including FAs, bioactive peptides, biogenic amines and oligosaccharides, which play critical roles in gut development [1,19,64]. Porcine milk replacers comprise mostly bovine milk components, often whey powder or skimmed milk powder, which are low in fat. The fat content of commercial milk replacers must correspond to the natural amount in sow milk and should be about 7 to 8% in the prepared liquid. In addition, the lipid profile should correspond to mature porcine milk. However, this is often not the case as the origin of the fat in milk replacers is mostly plant-based, often coconut oil and palm oil, which comprise a higher proportion of saturated and monounsaturated FAs than sow or bovine milk. For instance, coconut oil is rich in saturated lauric acid (C12:0, 49%) and contains in total about 82% saturated FAs, whereas palm oil comprises about 50% saturated FAs, 40% monounsaturated FAs and 10% PUFAs [65]. The introduction of creep feed has an effect on microbial activity, first suppressing fermentation in the hindgut but later stimulating the SCFA production when the microbes have adapted to the plant-based starchy components [66]. This can be assumed to be beneficial for the gut homeostasis. Moreover, how efficient the neonatal piglet can digest lipids in the milk replacer depends on the emulsifier used [67]. Albeit coming from a different species, bovine milk-derived emulsifiers (containing protein and phospholipids from milk fat globule membranes and extracellular vesicles) were shown to be superior to plant-based lecithin, such as soy lecithin, creating a more regular and smooth surface of fat droplets, which supports the emulsification process, lipase activity and fat absorption. When considering lipid sources for the development of milk replacers for piglets, also the positional distribution of FAs in triglycerides should be considered in terms of lipid digestibility [1].

Most pre-starter and starter diets are low in fat, comprising approximately 5% fat for improved flowability and pelleting of the diet. The fat in solid feeds mostly derives from cereals and/or processed vegetable oils, such as corn, soy, rapeseed or sunflower oils, due to economic reasons and availability. These vegetable oils contain large amounts of n-6 PUFAs and wide n-6:n-3 ratios; only rapeseed oil making an exception comprising higher amounts of n-3 PUFAs. The content of LA (C18:2 n-6) and ALA (C18:3 n-3) in rapeseed oil typically amounts to 20% and 10%, respectively, while about 8% FAs are saturated and 55% are monounsaturated oleic acid [68]. By contrast, sunflower oil is extremely rich in n-6 PUFAs, containing about 15% saturated and 85% unsaturated FAs, including monounsaturated FAs (14–43% oleic acid C18:1 n-9) and PUFAs (44–75% LA C18:3 n-3) [69] (Akkaya, 2018). Rapeseed oil exhibits good oxidative stability, which is higher than that of soybean and sunflower oils [68]. By having a lower oxidative stability and due to their lower availability and higher price, n-3 PUFA rich oils, such as linseed or hempseed oil, have been less used in (pre-)starter diets. The same is true for algae and seaweed [70], which are sources of long-chain n-3 PUFAs but have been less used due to their availability. However, this may change due to their beneficial impact on immunological processes, especially in the neonatal phase [18]. Aside from the type of PUFAs, selecting and combining sources of n-3 and n-6 PUFAs is important with regards to the synthesis of bioactive inflammatory lipid mediators as outlined below. Hemp seed oil, for instance, is described to have an optimal n-6:n-3 PUFA ratio of 3:1 for humans. Due to their close physiology, piglets may similarly benefit from a narrower n-6:n-3 PUFA ratio. In this respect, Yao et al. [57] demonstrated an improved immune status in piglets during the suckling period when the n-6:n-3 PUFA ratio of the diets for sows and piglets was adjusted to 9:1 by using a combination of n-6 PUFA rich corn oil and n-3 PUFA rich linseed oil.

### 4.3. Dietary n-3 PUFAs Show Anti-Inflammatory Effects in Suckling and Newly Weaned Piglets

Higher intake of n-3 and n-6 PUFAs via sow milk and solid diet of the piglets has been associated with increased incorporation of the FA into piglet enteric tissues and cell membranes. There, the PUFAs exert their bioactivity for immune responses and the epithelial barrier function [1]. Most research evidence for the anti-inflammatory capacity in piglets exists for n-3 PUFA-rich oils such as linseed oil and fish oil, which seem to be helpful in transition periods such as weaning when piglets are prone to intestinal inflammation and immune suppression [1,7,17,71]. Table 2 summarizes the effects of n-3 PUFA and MCFA-rich oils on the physiology of suckling and newly weaned piglets from studies published for 2017 to 2021. Beneficial effects of n-3 PUFAs when provided to reproducing sows and piglets have been published before and can be found in respective reviews, for instance of Tange and de Smet [28]. When comparing older with more recent investigations about the effects of FAs on piglet’s physiology, it should be kept in mind that pig breeds and lines have changed in terms of their reproductive performance (increased litter size) over the past ten years, which may have increased the demand of the sows for fat and essential FA due to the increased litter sizes and the consequent higher delivery of essential fatty acids during foetal development and production of colostrum and milk. Especially, the greater number of small size piglets may benefit from sow colostrum and milk enriched with n-3 PUFAs and fat in general. Likewise, plant genetics have changed, increasing the yield from seeds. Not only the PUFA concentration but also the advancements in the production technology of vegetable oils may change PUFA profiles as well as the amount of chemical residuals from extraction processes from refineries [72]. The latter may have an impact on study results as they may act pro-inflammatory.

Feeding n-3 PUFAs from early life as a dietary supplement may be challenging because the uptake of food aside from sow milk can vary substantially within a litter. Since the dietary PUFAs are transferred into sow colostrum and milk, manipulating the dietary PUFAs composition of sow’s late gestation and lactation diets may be the most efficient strategy to increase the n-3 PUFAs intake of piglets before the intake of creep feed or solid feed increases in the week before weaning [18]. Of further note, more n-3 PUFAs reach the piglet through colostrum and mature milk than via the placenta [18,76], further justifying the fortification of sow’s late gestation and lactation diets with n-3 PUFAs.

Consistent with earlier studies, the effects of n-3 PUFAs or MCFA supplementation on sow performance and litter physiology were ambiguous, depending on the type of the oil used, supplementation level, balancing of the dietary fat and energy level, meeting of energy requirements, composition of the basal and control diets and the lipid source used in the control diet. Overall, n-3 PUFAs supplementation seemed to have a certain beneficial effect on piglet survival during the suckling phase [7,62]. This may have been due to an improved supply with immunoglobulins via colostrum and milk, enforcing the passive immunity of the piglets during the suckling phase [45,71]. Results from the literature generally showed increased n-3 PUFA levels in piglet’s plasma, confirming an improved supply with essential FA when provided via sow’s milk or in piglet’s diet pre- and postweaning [17,18,71]. Moreover, effects of the dietary lipid source were generally found if the supplementation level was not too low. For instance, McAfee et al. [17] only observed effects when a fish oil product was added at 1% and used this supplementation level in their second experiment, whereas an inclusion level of 0.25 and 0.5% proved less promising. These findings were supported by the earlier results of de Quelen et al. [59] who found that extruded linseed oil in the sow diet increased the n-3 PUFA status of the piglet in the suckling phase as indicated by higher levels in the liver and brain. However, a minimal content of linseed oil (3% compared to 1.5%) and hence C18:3 n-3 seemed to be necessary to reach this effect.

Long-chain n-3 PUFAs in fish oil supported the gut health and function of suckling and weaned pigs, which incorporate more EPA and DHA into membranal phospholipids including those in the gut [1]. However, there is some controversy as a diet rich in C18:3 n-3 PUFA fed during pregnancy and lactation increased jejunal permeability in suckling piglets at weaning, as shown previously [59]. The diverging findings show the need to clarify the mechanisms of lipid signalling and uptake along the gut of suckling piglets. Most n-3 PUFA rich vegetable oils such as linseed, hempseed and algae oil mainly contain ALA and not the longer chained EPA and DHA, which show bioactivity. The elongation steps involve several desaturases and thus are saturable processes. Therefore, enzyme activities may be limited, preventing clear effects. In addition, no data on the peroxidation status of the oil was provided in the study of de Quelen et al. [59], which, if high, may have increased the oxidative stress level in the gut, thereby compromising the intestinal integrity. By contrast, postweaning data for piglets indicated that an addition of 5% linseed oil was effective in enhancing intestinal integrity and barrier function, which was due to a modulation of necroptosis and TLR4/NOD signalling pathways [74]. Using a commercial product in their study, McAfee et al. [17] missed providing information about the nature of the fish oil and its production process, similar to other studies not providing a detailed description of the lipid source used [75]. However, this would have provided valid information in terms of the bioactivity of the fish oil.

There has been some research interest on MCFA as well, whereby coconut oil and palm oil are the best investigated natural lipid sources of MCFA. Due to its high lauric acid content (C12:0), coconut oil exerts antimicrobial, antiviral and antifungal properties [76,77]. Due to similar reasons, palm oil may show similar microbiota-modulating properties [78]. Effects of MCFA on the inflammatory response were inconclusive but may be related to the individual MCFA source and hence FA composition added to the diet. The site of esterification of the respective FA in the triglyceride molecule is also important, influencing the efficacy of the supplemented lipid. In this respect, Ferrara et al. [73] did not find an effect of a MCFA mix of 50% of caprylic and capric acid on the jejunal histo-morphology and immunology, including intraepithelial lymphocytes (IEL), CD3-positive IEL and CD2-, CD5-, CD8β-, CD16-and γδ TCR-positive IEL, when compared to the control. Unfortunately, the source of the fat used in the control diet was not provided, impeding a comparison of the FA composition between control and test diets. By contrast, Chen et al. [71] found a beneficial effect of 0.775% MCFA on colonic expression of genes related to barrier function, inflammation and diarrhoea incidence. Additionally, in their study, the authors failed to provide the information on the type of product used and the actual (analysed) FA composition.

Often n-6 PUFA-rich soybean oil was used as the control/reference oil [7,17,71]. Others used corn oil [42], another highly available and economically attractive n-6 PUFA-rich lipid source. The utilization of a similar reference oil has the advantage of allowing the comparison of results among studies. However, corn oil has a wide n-6:n-3 PUFA ratio; hence, results may look different when oils with narrower n-6:n-3 PUFA ratios or oils comprising low PUFA levels; mostly monounsaturated and saturated FA, are used as control. Saturated long-chain FAs such as in tallow may act as an anti-inflammatory as they are very stable against autoxidation. However, information is scarce towards the effects of animal-based saturated fat sources on gut integrity and function in nursery piglets under practical conditions.

Since PUFAs, irrespective of the site of the double bonds, are prone to autoxidize, authors should be encouraged to provide information on how they controlled the autoxidation process of their PUFA-rich oils and provide the respective (analysed) data for their feed in future publications. For instance, cold-pressed linseed oil is very susceptible to autoxidation, reducing its shelf life, and should be kept refrigerated until consumption. Refrigeration of sow’s gestation and lactation diets is mostly not feasible under practical conditions. Oxidized FAs may annihilate effects and actually harm the animal by systematically increasing the oxidative stress load in the intestines. The importance to control the autoxidation of the lipids used was indicated by the study of Su et al. [42] (please see Section 3.4). Taken together, to enable better comparison of results among studies on dietary PUFAs, it may be recommended to provide details on the lipid sources used (control and test lipid source) and the analysed FA composition as well as information on the peroxidation of the utilized lipids and diets at the start and end of the experiment.

## 5. Microbially-Derived FA

In addition to altering the structure and function of cell membranes, n-3 PUFAs including DHA, EPA, ALA and docosapentaenoic acid modulate the intestinal immune tolerance by influencing the gut microbiota composition [79]. Simultaneously, the gut microbiota is a source of FA [4]. Microbes produce a range of FAs, including straight- and branched-chained SCFAs, MCFAs and long-chain FAs during fermentation. Moreover, evidence is increasing that gut bacteria in monogastric animals generate unique bioactive lipid mediators from dietary lipids and during fermentation including conjugated, hydroxyl- and oxo-FAs which seem to have anti-inflammatory activity [4,80,81]. While there has been substantial research interest on these microbial lipid derivatives in ruminants, in pigs the focus has been on the benefits of SCFAs on gut and systemic inflammation and health. Most evidence for intestinal SCFA production in piglets exists from the fourth week of life onwards [82,83], while only recent research has started to have a closer look into the neonatal phase [66,84]. Accumulating research data show that physiologically relevant concentrations of SCFA exist in the ileum, cecum and colon from the first days of life and rise until weaning [66,85]. Already the meconium comprises small amounts of SCFAs, which may play an important role for host mucosal nutrition and priming in neonatal piglets [86,87]. The SCFA profiles and amounts fluctuate postnatally, mirroring maturational changes in microbial species abundances [65,84,85]. To illustrate this, SCFAs increased from day 2 to 13 of life, while their concentration decreased again until day 20 [66].

Due to its beneficial action on intestinal integrity and functioning, special research interest has been on dietary formulations to increase the intestinal concentration of butyrate [88], while other classes of microbial FA derivatives such as conjugated LA, and to a lesser degree hydroxy FAs, and oxo FAs have been of research interest in ruminants and in the production of fermented milk products (as reviewed by [4]). These specific FA derivatives are not generated by mammalian cells, being a feature uniquely linked to the intestinal microbes. Various bioactivities have been described for these FA derivatives in humans [4]. Bioactivities relevant for early piglet development and health are related to their anti-inflammatory and cytoprotective effects, effects on lipid/energy metabolism, and protection of the gut barrier function. These are mediated by transcriptional regulation, e.g., conjugated LA isomers have been shown to modify insulin sensitivity via activation of PPARs and liver X receptorα (LXR-α), and act via suppression of macrophage activity, thereby interrupting inflammatory processes [4]. Oxo-FAs, in turn, are involved in appetite regulation by stimulating the secretion of cholecystokinin via G-protein receptor (GPR) activation, possibly GPR-40 [80]. Although these metabolic processes have been first described for the rumen, taxonomic and functional microbiome data for pig intestinal digesta show similar species abundances [89,90]. Bacteria including *Butyrivibrio*, *Lactobacillus*, and *Megasphaera* biohydrogenate LA and ALA, thereby producing their conjugated forms, which are isomers with conjugated double bounds [4]. Especially, lactobacilli belong to early colonizers in piglets and are predominant in the upper digestive tract throughout the suckling phase and beyond [84,91]. Beneficial effects of conjugated LA on the immune status of piglets have been described in the literature. Especially, the supplementation of sow’s gestation and lactation diets with, for instance, 2% conjugated LA may be a practical strategy for enhancing passive immune transfer and improving the immune status and overall gut health of nursery piglets with an immune-stimulating carry-over effect until postweaning [92]. Moreover, supplementing sow’s lactation diet with 1% conjugated LA improved milk yield without affecting the sow’s body weight, as was the case with soybean oil supplementation [93,94]. Against this background, intestinal production of conjugated LA and other FA derivatives by the gut microbiota may be thinkable, even in very young piglets. In this respect, we recently found increased long-chain FA concentrations including C18:1 n-9, C18:2 n-6, and C18:3 n-3 in the cecal and colonic digesta of 10- to 11-week-old pigs, which were, at least partially, attributable to microbial FA synthesis [31].

## 6. Conclusions and Perspectives

The literature summarized in this review indicates the role of dietary lipids for the development and growth of new-born piglets which goes beyond the preweaning phase. Adequate nutritional supply of bioactive FAs, such as n-3 rich PUFAs, may control the inflammatory response and support intestinal barrier functions by modifying the FA profile in membranous phospholipids when provided throughout the suckling phase. Manipulation of the FA composition of colostrum and milk by sow’s late gestation and lactation diets proved to be one efficient strategy to increase a piglet’s n-3 PUFA intake. Advances in lipidomics technology and cell culture models have enabled the characterization of novel lipid mediators involved in the epithelial inflammatory signalling of long-chain PUFAs, such as AA, EPA and DHA. Moreover, gut mucosal FA receptors sense the abundance of SCFAs, MCFAs and long-chain FAs and trigger a mucosal response before the FAs are absorbed and built into membranes, thereby influencing glucose metabolism, satiety and inflammatory responses. Functional data on these receptors and their gut region-specific development are still scarce for new-born piglets. Moreover, little has been described in the literature about long-chain FAs derived by the gut microbiota during the neonatal period. Due the important role that the gut microbial maturation plays at this age, the identification and the characterization of their (anti-)inflammatory capacity of microbial FA metabolites in the neonatal porcine gut should receive more attention in future studies. Although the superiority of n-3 PUFAs compared to n-6 PUFAs in terms of their anti-inflammatory signalling has been shown, both types of PUFA are prone to autoxidation, which actually increases the oxidative damage to the gut lining. Therefore, certain care in the storage of complete feeds supplemented with n-3 PUFA oils is recommended. Moreover, a certain standardization of experiments and provision of information on products used may be advised. This includes a detailed description of the lipid sources used (control and test lipid source), the analysed dietary FA composition, as well as information on the oxidative stability of the lipids and complete diets at the start and end of the experiment.

## Figures and Tables

**Table 1 animals-11-02781-t001:** Average fatty acid composition of porcine colostrum and transient-mature milk.

Item ^1^	Luise et al. [48] ^2^	Aguinaga et al. [49] ^3^		Laws et al. [47] ^4^		Vodolaszka and Lauridsen [18] ^5^	
	Colostrum	Colostrum	Milk	Colostrum	Milk	Colostrum	Milk
Total fat (%)	3.30	3.76	5.91	NA	8.35	NA	NA
Fatty acid (%)							
C14:0	2.48	2.11	3.06	0.91	2.82	1.28	3.42
C14:1	ND	ND	ND	0.03	0.14	ND	ND
C16:0	27.37	26.50	31.20	22.75	31.35	21.77	29.07
C16:1n-9	1.04	ND	ND	ND	ND	1.18	0.34
C16:1n-7	3.83	3.70	7.67	3.36	8.31	2.75	7.89
C17:0	0.36	ND	ND	ND	ND	ND	ND
C17:1	0.30	ND	ND	ND	ND	ND	ND
C18:0	4.34	5.54	4.61	5.57	4.91	5.04	3.95
C18:1trans	0.35	ND	ND	ND	ND	ND	ND
C18:1n-9	28.72	37.00	37.20	35.75	33.43	31.50	22.55
C18:1n-7	ND	ND	ND	ND	ND	ND	ND
C18:1n-11	3.04	ND	ND	ND	ND	2.85	1.91
C18:2n-6	22.69	16.70	9.57	25.15	15.28	28.03	22.67
C18:3n-6	0.30	ND	ND	0.31	0.04	0.79	0.45
C18:3n-3	1.47	1.06	0.64	1.58	1.10	3.19	3.16
C18:4n-3	ND	ND	ND	ND	ND	0.12	0.08
C18:2n-9 trans-11	0.08	ND	ND	ND	ND	ND	ND
C20:0	0.06	ND	ND	ND	ND	ND	ND
C20:1n-9	ND	ND	ND	0.27	0.34	ND	ND
C20:2n-6	0.38	0.20	0.09	0.50	0.35	0.46	0.37
C20:3n-6	ND	ND	ND	ND	ND	0.83	0.56
C20:4n-6	1.18	0.90	0.45	1.04	0.60	0.16	0.13
C20:5n-3	ND	0.10	0.06	0.25	0.38	0.12	0.09
C22:1n-9	ND	ND	ND	0.11	0.10	ND	ND
C22:2n-6	ND	ND	ND	0.25	0.01	ND	ND
C22:4n-6	0.43	ND	ND	ND	ND	ND	ND
C22:5n-6	ND	ND	ND	ND	ND	0.13	0.11
C22:5n-3	ND	0.29	0.12	0.39	0.24	0.41	0.27
C22:6n-3	0.37	0.04	0.02	0.33	0.16	0.06	0.02
SFA	34.26	35.40	40.10	30.00	39.53	26.62	38.40
MUFA	34.23	44.50	48.48	39.60	42.35	38.60	31.68
MUFA, n-9	29.76	37.60	38.16	36.13	33.86	32.68	22.89
PUFA	26.92	19.29	10.94	30.35	18.54	33.77	28.27
PUFA:SFA	0.79	0.55	0.27	1.01	0.48	1.27	0.74
PUFA, n-6	24.99	18.60	10.58	28.00	16.57	29.55	24.27
PUFA, n-3	1.84	1.57	0.88	2.67	1.91	4.19	4.17
n-6:n-3 ratio	13.56	11.85	11.97	11.20	8.98	7.65	6.45

^1^ SFA, saturated fatty acids; MUFA, mono-unsaturated fatty acids, PUFA, polyunsaturated fatty acids. ND, not detected; NA, not applicable as data were not provided. Values for total fat and fatty acids are averaged across dietary treatments. ^2^ Sow breeds: Italian Duroc, Italian Landrace and Italian Large White. Colostrum collected on farrowing day. ^3^ Sow breed: Iberian. Colostrum collected on farrowing day; transient to mature milk averaged for days 5, 12, 19, 24, 36 of lactation. ^4^ Sow breed: commercial genotype: 25% Meishan, 12.5% Duroc, 62.5% Large White × Landrace. Colostrum collected on farrowing day; transient to mature milk averaged for days 3, 7, 14, 21 of lactation. ^5^ Sow breed: Landrace × Yorkshire. Colostrum averaged for days 0 and 2; transient to mature milk averaged for days 16 and 28 of lactation.

**Table 2 animals-11-02781-t002:** Effects of dietary lipids on intestinal physiology and inflammatory response in piglets when lipids were provided in sow gestation and lactation diets and/or piglet feed ^1^.

Piglet Age	Diet Type	Main Dietary Ingredients	Lipid Type	Supplementation Level (%)	Effect on Milk Composition	Effects on Piglet Physiology	Effect on Piglet Growth	Effect on Sow Performance	Reference
Suckling, day 1 to day 21	Sow gestation diet from day 108 of gestation. Lactation diet	Corn, soybean meal	Coconut oil	3	colostrum + milk: saturated FA ↑; no effect on IgG, IgA, IgM	piglets: plasma C14:0, IgG, IgA ↑	no effect on litter ADG + BW + survival rate	no effect on BW, ADFI, backfat thickness	[46]
			Palm oil	3	colostrum + milk: saturated FA ↑	piglets: plasma C16:0, IgG, IgA ↑			
			Soybean oil	3	colostrum + milk: fat, n-6 PUFA, n-3 PUFA ↑	piglets: plasma IgG, IgA ↑			
			Mixed oil (Coconut oil, palm oil soybean oil)	3 (1% of each)	colostrum + milk: fat ↑	piglets: plasma IgG, IgA ↑			
Postweaning, from day 28 of life	Weaner diet (from 28 days of life)	Commercial starter diet	Short-chain organic acids (SOA, fumaric acid 0.41%, lactic acid 0.32%)	1.05	NA	CD2−/cd+ ↑	NA	NA	[73]
			MCFA (caprylic acid 50%, capric acid 50% + SOA)	MCFA 0.3, SOA 1.05	NA	NA	NA	NA	
Postweaning, experimental period 21 days	Starter diet	Corn, soybean meal	corn oil	5	NA	NA	NA	NA	[74]
			linseed oil	5	NA	intestinal ALA, EPA + total n-3 PUFAs, morphology, jejunal lactase activity, and claudin-1 protein expression ↑; intestinal expression ofTLR, 4MyD88NF−κB, NOD1, NOD2, RIPK2 ↓	NA	NA	
Suckling, day 1 to weaning day 22 of life	sow gestation diet from day 85 of gestation, sow lactation diet	Corn, barley soybean meal, expanded soybean	coated sodium butyrate	0.1	fat, protein, IgA, IgG, IgM ↑	survival rate of suckling piglets, serum IgA, IgG, IgM, colonic expression of CDLN1, ZO1 ↑; diarrhea incidence, crypt depth, colonic expression of TLR4, MγD88, IL6, IL10, IL1B, TNFA ↓; no effect on OCLN	no effect on growth, survival rate of piglets ↑	weaning-to-estrus interval of sows ↓	[71]
			MCFA (MCT)	0.775	fat, protein, IgA, IgG, IgM ↑	colonic expression of CDLN1, ZO1 ↑; diarrhea incidence, crypt depth, colonic expression of TLR4, MγD88, IL6, IL10, IL1B, TNFA â; no effect on OCLN	no effect	weaning-to-estrus interval of sows ↓	
			n-3 PUFA (ALA, DHA, EPA)	6.82	fat, protein, IgA, IgG, IgM ↑	colonic expression of CDLN1, ZO1 ↑; diarrhea incidence, crypt depth, colonic expression of TLR4, MγD88, IL6, IL10, IL1B, TNFA, NFKB â; no effect on OCLN	no effect	weaning-to-estrus interval of sows ↓	
Suckling, day 1 to 28	Sow gestation diet from day 105 of gestation, lactation diet	Wheat, corn, soybean	salmon oil (control: soybean oil) + 2 levels: 1.79 versus 5.98%	1.79 versus 5.98%	colostrum + milk: n-3 PUFA C20:5, C22:5, C22:6 á; n-6 PUFA C18:2 ↓	at weaning: adipose tissue + plasma: total n-3 PUFA ↑; total n-6 PUFA plasma â; higher level caused stronger effect	litter: no effect on ADG + BW + ADFI; energy intake ↑	no effect on backfat thickness + BCS	[16]
Weaning (day 22) to day 42 postweaning	starter diet	Corn, wheat, soybean meal	microalgae	3.12	NA	plasma EPA, DHA, total n-3 PUFA ↑; serum IL-1β, IL-6, TNF-α ↓	no effect on BW + ADG	NA	[75]
			fish oil	1.25	NA	plasma EPA, DHA, total n-3 PUFA ↑; serum IL-1β, IL-6, TNF-α, acute-phase-response post LPS-challenge ↓	no effect on BW + ADG	NA	
Day 1 to 3 weeks postweaning (day 51 of life)	Sow late gestation diet from day 101 of gestation, lactation diet until day 16 of lactation; nursery diet from 1 week prior to 3 weeks after weaning, weaning on day 31 day of life	Corn, dehulled soybean meal	fish oil product (39.2% fat (by acid hydrolysis) with EPA + DHA making up 13.8% and 11.4% of total fat)	0.25	no effect	NA	NA	NA	[17]
				0.5	no effect	NA	NA	NA	
				1.0	colostrum DHA ↑	Colostrum DHA ↑; piglet: body weight preweaning, plasma n-3:n-6 PUFA ratio at postweaning day 1+3 ↑; cortisol, corticosteroid-binding globulin, haptoglobin postweaning day 1 + 3 ↓	BW on day 3 postweaning ↑	NA	
Suckling, day 1 to 18	Sow late gestation diet from day 110 of gestation, lactation diet	Corn, soybean meal	lysophospholipids (lysophosphatidylcholine, lysophosphatidylinositol, lysophosphatidylethanolamine, & lysophosphatidic acid)	0.05	no effect on colostrum or day 18-milk	Sow: milk saturated FA â; milk unsaturated FA ↑ piglet: mortality â; litter size & weight day 9 & 18, crypt cell proliferation ↑;	day 9: Litter size, litter weight ↑; day 18: Litter size, litter weight ↑; mortality â	no effect on BW, backfat thickness	[62]
Suckling, day 1 to 26	Sow gestation diet from day 28 of gestation, lactation diet	Corn, barley, wheat bran, distiller grains	low n-6:n-3 PUFA ratio (sources: soybean oil, linseed oil)	control: 13:1 gestation, 10:1 lactation; low ratio: 4:1 gestation & lactation	ALA, ETA, EPA in end lactation-milk ↑, DHA, DGLA, DLA in colostrum, n-6:n-3 ratio in colostrum + milk ↓	Colostrum & milk: n-6:n-3 PUFA â; Litter weight gain postweaning ↑; no effect on oxidative status	litter + piglet: ADG birth to weaning ↑; survival rate ↑	no effect on sow body weight, weight gain and loss during gestation and lactation; piglets born alive ↓	[7]
Suckling, day 1 to 28	Sow late gestation diet from day 108 of gestation, lactation diet	Barley, wheat, soybean meal	Hemp seed oil (control: soybean oil)	5	colostrum + milk + sow plasma: n-3 PUFA ↑; n-6:n-3 ↓; effect stronger with only hempseed oil; colostrum + milk: no effect on IgA, IgG, IgM	piglet plasma: total n-3 PUFA, C18:3 n-3,C20:3 n-6, C20:5 n-3, C22:5 n-3 ↑; n-6:n-3 ↓; effect stronger with only hempseed oil; hempseed oil: plasma IgA ↓; day 28; no significant effect on COX-2, IL-10 and TNF-α expression along proximal, mid, distal small intestine	BW in first week, survival rate birth to weaning ↑	no effect on BW; still born piglets ↓	[18]
			Hemp seed oil:soybean oil mix (50:50)	2.5 hempseed oil:2.5 soybean oil			no effect on BW	no effect on BW; total born + live born ↑; still born piglets ↓	

^1^ FA, fatty acids; MCFA, medium-chain fatty acids; PUFA, polyunsaturated fatty acids; Ig immunoglobulins; MCT, medium-chain triglycerides; ALA, α-linolenic acid; ETA, eicosatrienoic acid; EPA, eicosapentaenoic acid; DHA, docosahexaenoic acid; DGLA, dihomo-γ-linolenic acid; LPS, lipopolysaccharide; NA, not applicable.

## Data Availability

Data sharing is not applicable to this article.

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
