# Peer review of "The Role of Dietary and Microbial Fatty Acids in the Control of Inflammation in Neonatal Piglets"

_animals, 2021, doi:10.3390/ani11102781_

Round 1

Reviewer 1 Report

Dear Author

I enjoyed reading your work, which I consider a good contribution to the scientific field. I leave my suggestions that aim to enhance the clarity of it.

Comments:

  1. The author did not mention that during the gut maturation the regulatory role of the enteric nervous system is of key importance. It looks like she completely forgot about the roles of ENS. Moreover, it has been shown that fatty acids modulate enteric neurons activity (Soret R, Chevalier J, De Coppet P, Poupeau G, Derkinderen P, Segain JP, Neunlist M. Short-chain fatty acids regulate the enteric neurons and control gastrointestinal motility in rats. Gastroenterology. 2010 May;138(5):1772-82.). I strongly advise to discuss this issue in the manuscript.
  2. The porcine colostrum has been shown to influence the gastrointestinal structure including the activity of ENS (WoliÅ„ski J, SÅ‚upecka M, Weström B, Prykhodko O, Ochniewicz P, Arciszewski M, Ekblad E, Szwiec K, Ushakova, Skibo G, Kovalenko T, Osadchenko I, Goncharova K, Botermans J, Pierzynowski S. Effect of feeding colostrum versus exogenous immunoglobulin G on gastrointestinal structure and enteric nervous system in newborn pigs. J Anim Sci. 2012 Dec;90 Suppl 4:327-30.) This work should be included in the text and discussed.
  3. In Chapter 6, I actually don't see any clear-cut conclusions. I would suggest the author to include very unambiguous conclusions that would reflect the significance of this manuscript.

Author Response

Comments:

  1. The author did not mention that during the gut maturation the regulatory role of the enteric nervous system is of key importance. It looks like she completely forgot about the roles of ENS. Moreover, it has been shown that fatty acids modulate enteric neurons activity (Soret R, Chevalier J, De Coppet P, Poupeau G, Derkinderen P, Segain JP, Neunlist M. Short-chain fatty acids regulate the enteric neurons and control gastrointestinal motility in rats. Gastroenterology. 2010 May;138(5):1772-82.). I strongly advise to discuss this issue in the manuscript.
  2.  

BM: Thank you for this suggestion. The role for neuronal signaling was mentioned and the paper cited (New Lines 238-241).

  1. The porcine colostrum has been shown to influence the gastrointestinal structure including the activity of ENS (WoliÅ„ski J, SÅ‚upecka M, Weström B, Prykhodko O, Ochniewicz P, Arciszewski M, Ekblad E, Szwiec K, Ushakova, Skibo G, Kovalenko T, Osadchenko I, Goncharova K, Botermans J, Pierzynowski S. Effect of feeding colostrum versus exogenous immunoglobulin G on gastrointestinal structure and enteric nervous system in newborn pigs. J Anim Sci. 2012 Dec;90 Suppl 4:327-30.) This work should be included in the text and discussed.

BM: Thank you. Because the aim of this review is not to discuss the effect of colostrum in general, this paper does not really fit into this review.

  1. In Chapter 6, I actually don't see any clear-cut conclusions. I would suggest the author to include very unambiguous conclusions that would reflect the significance of this manuscript.

BM: the conclusion section was modified.

Reviewer 2 Report

This is an excellent review (my congratulations to Barbara), which merits to be published and I am sure will receive the interest of the readers of the journal

A minor change should be done in line 5 of the chapter titled "Role of lipids for piglet’s development" to change the word "fecal" by prenatal?

Author Response

BM: Thank you very much for your judgement of the quality of my review manuscript. The change has been done accordingly (New Line 110).